# Is It Time for a Reset in Arctic Governance?

**Oran R. Young** 

Bren School of Environmental Science and Management, University of California (Santa Barbara), Santa Barbara, CA 93103, USA; oran.young@gmail.com or young@bren.ucsb.edu

**Abstract:** Conditions in the Arctic today differ from those prevailing during the 1990s in ways that have far-reaching implications for the architecture of Arctic governance. What was once a peripheral region regarded as a zone of peace has turned into ground zero for climate change on a global scale and a scene of geopolitical maneuvering in which Russia is flexing its muscles as a resurgent great power, China is launching economic initiatives, and the United States is reacting defensively as an embattled but still potent hegemon. This article explores the consequences of these developments for Arctic governance and specifically for the role of the Arctic Council. The article canvasses options for adjusting the council's membership and its substantive remit. It pays particular attention to opportunities for the council to play a role in managing the increasingly complex Arctic regime complex.

**Keywords:** climate change; geopolitics; governance; regime complex

## 1. Introduction

The architecture of Arctic governance, centered on the role of the Arctic Council treated as a "high-level forum" designed to promote "cooperation, coordination and interaction among the Arctic states," reflects conditions prevailing in the 1990s, a period marked by the end of the cold war, the collapse of the Soviet Union, and the reduction in tension in the High North that followed these events [1]. Most analysts agree that the Arctic Council has performed well during the 25 years that have elapsed since its establishment under the terms of the 1996 Ottawa Declaration on the Establishment of the Arctic Council. The accomplishments of the council have exceeded the expectations of most of us who participated in the processes eventuating in the adoption of the Ministerial Declaration on 19 September 1996. In a moment of enthusiasm, the Arctic Council Ministerial Meeting held in Kiruna, Sweden on 15 May 2013 at the close of the Swedish chairmanship went so far as to adopt a statement declaring that " . . . the Arctic Council has become the pre-eminent high-level forum of the Arctic region and we have made this region into an area of unique international cooperation" [2]. In support of this expansive assertion, the ministers proclaimed that: "We have achieved mutual understanding and trust, addressed issues of common concern, strengthened our co-operation, influenced international action, established a standing secretariat and, under the auspices of the Council, Arctic States have concluded legally binding agreements. We have also documented the importance of science and traditional knowledge for understanding our region and for informed decision-making in the Arctic" [2]. Even discounting for a normal dose of inflated rhetoric, this statement reflects a remarkable degree of confidence regarding the achievements of an institution rooted in nothing more than the provisions of a ministerial declaration.

Yet, conditions in the Arctic today differ dramatically from those prevailing at the time of the creation of the Arctic Council [3]. In the 1990s, the Arctic was a peripheral region, no longer critical as the front line of the cold war and not yet regarded as a central focus among those concerned with the threat of climate change or the prospects for extracting raw materials needed to supply advanced

industrial systems. For the most part, non-Arctic states did not object to claims on the part of the Arctic states to dominance in the Arctic arising from their self-proclaimed "sovereignty, sovereign rights, and jurisdiction" in the region [4]. So long as Arctic concerns did not spill over into global arenas, in other words, the rest of the world saw no harm in leaving the affairs of the Arctic to the Arctic states.

In recent years, however, the Arctic has moved from the periphery to the center with regard to matters of global concern. The effects of climate change are surfacing more rapidly in the Arctic than in any other part of the Earth system. Due to the operation of feedback mechanisms involving the recession and thinning of sea ice, the melting of permafrost, and shifts in the mass balance of the Greenland ice sheet, these effects of climate change are producing dramatic consequences on a global scale. In addition, the increasing accessibility of the Arctic has generated a surge of interest in extracting the Arctic's natural resources, and especially, its world-class reserves of oil and natural gas. To take a single example, the extraction of natural gas on Russia's Yamal Peninsula and its shipment to Asian and European markets using specially designed LNG tankers has become a focus of attention in the financial capitals of the world. Some see the Arctic as an arena that will attract investments on the order of $1 US trillion during the coming years [5].

Broader geopolitical shifts have served to heighten the significance of these developments. A resurgent Russia has taken steps to reclaim its status as a great power, articulating renewed claims to a leading role in the affairs of the Far North and strengthening military assets in the Russian Arctic. China, as an emerging superpower, has taken steps to include the Arctic within the scope of its Belt and Road Initiative, leading among other things, to a series of bilateral moves relating to economic investments or proposed investments in Russia, Finland, Iceland, Greenland, and Canada. The United States has become sensitive about its superpower status and begun to treat the Arctic activities of others as hostile initiatives threatening America's interests. All these developments are playing out within a shifting global context characterized by the decline of the postwar world order but, so far at least, a lack of clarity regarding the global order of the future. The relative tranquillity associated with the Arctic's peripheral status has come to an end. Whereas an interest in the Arctic once seemed quixotic to students of international politics, not a week goes by today without one or more international conferences in which numerous people advance speculative projections regarding the future of the Arctic.

It, therefore, makes sense to ask whether the architecture of Arctic governance, put in place during the 1990s, is well-suited to address needs for governance arising under the conditions prevailing in the 2020s. Can we make incremental and pragmatic adjustments in the existing arrangements to accommodate changing conditions? Will we find ourselves faced with a need to consider changes in the existing architecture that are constitutive in character? Will the new conditions prevailing in the Arctic impede all efforts to adjust or restructure existing governance arrangements? I set forth my responses to these questions in three steps. The next section describes and clarifies the premises and practices that define the architecture of Arctic governance put in place during the 1990s. The following section then asks whether these premises and practices can provide a viable platform for coming to terms with needs for governance regarding Arctic issues in the 2020s. The final substantive section then turns to the issue of institutional reconfiguration, considering innovations designed to meet the needs of the 2020s, with particular reference to the role of the Arctic Council in what is often described as the 'new' Arctic [6].

## 2. The Existing Architecture: Premises and Practices

We normally look to the provisions of formal documents, like the 1996 Ottawa Declaration, in identifying the principal elements of governance systems. But it is important to recognize that these systems rest on fundamental premises or assumptions that provide the analytic context for specific institutional arrangements, even though they may remain unstated in formal documents. The Arctic is no exception in these terms. Three major premises are embedded in the thinking of those who formulated the specific provisions of the Ottawa Declaration.

*The Arctic is a distinctive, low-tension international region with a policy agenda of its own centered on issues of environmental protection and sustainable development.* It is not self-evident that the circumpolar Arctic constitutes a distinctive international region that makes sense in thinking about matters of public policy. Yet during the mid-1980s, a group of policy analysts began to make the case for treating the Arctic as a distinctive region [7]. This effort received powerful support in Mikhail Gorbachev's 1 October 1987 Murmansk speech in which the then president of the Soviet Union said "[l]et the North of the globe, the Arctic, become a zone of peace," called for international cooperation to pursue this goal, and proposed a suite of cooperative initiatives dealing with arms control, economic development, the opening of the Northern Sea Route, environmental protection, and scientific research [8]. Brian Mulroney, then prime minister of Canada, added momentum in a speech in Leningrad (now St. Petersburg) on 24 November 1989 in which he said: " ... and why not a council of Arctic countries eventually coming into existence to coordinate and promote cooperation among them" [9]. This set the stage for the so-called Finnish Initiative bringing together the eight Arctic states in a collaborative effort leading to the 14 June 1991 Rovaniemi Declaration on the Protection of the Arctic Environment launching " ... a joint Action Plan of the Arctic Environmental Protection Strategy" [10,11]. This declaration established the practice of treating the circumpolar Arctic as a distinctive international region and paved the way for the 19 September 1996 Ottawa Declaration on the Establishment of the Arctic Council, not only calling on the Arctic states to cooperate and coordinate on "issues of sustainable development and environmental protection in the Arctic" but also (famously) specifying that the "Arctic Council should not deal with matters related to military security" [1].

*The interests of the Arctic states are paramount when it comes to addressing needs for governance in the region.* In the process, the Arctic states made it clear that they could and should control the international relations of the Arctic region by virtue not only of their de facto control but also of their de jure authority in the region. There is an internal debate that surfaces from time to time regarding the division of authority and control between the five Arctic Ocean coastal states (Canada, Denmark, Norway, Russia, and the United States) and the eight Arctic states that are members of the Arctic Council (the five plus Finland, Iceland, and Sweden). Most recently, the Arctic five have taken the lead in negotiating the terms of the 2018 Central Arctic Ocean fisheries agreement. Nevertheless, all the Arctic states stand together when it comes to matters regarding the primacy of the Arctic states in addressing issues pertinent to Arctic governance. In justifying this stand, the Arctic states can point to their jurisdiction over Arctic lands, to general provisions of the law of the sea, and to several specific provisions of the 1982 UN Convention on the Law of the Sea dealing with the role of coastal states relating to maritime governance. Ultimately, however, the claims of the Arctic states concerning the primacy of their interests in addressing matters of Arctic governance rest on the political realities of the region.

*The Arctic is not a vacuum when it comes to arrangements required to address specific needs for governance.* Given the tendency of outsiders to compare the Arctic and the Antarctic and to conclude that there is a need for an Arctic Treaty analogous to the 1959 Antarctic Treaty, it is understandable that the Arctic states have made a concerted effort to demonstrate that the Arctic is not the "wild west" regarding matters of governance [12]. In fact, they argue, the Arctic already has an extensive system of governance, dating back to the 1920 Spitsbergen Treaty and the 1973 Agreement on the Conservation of Polar Bears. The terrestrial parts of the region are subject to the domestic laws of the individual Arctic states. But even in the case of the marine parts of the region, there are applicable constitutive provisions of the law of the sea coupled with a raft of specific arrangements dealing with matters like commercial shipping, fishing, and marine mammals. The role of the Arctic Council, on this account, is not to assume responsibility for some functionally defined set of issues but to provide a policy forum for the Arctic states to consider matters of coordination relating to this complex web of governance arrangements. This explains why the Arctic states launched the council through a simple ministerial declaration rather than through the negotiation of an international legally binding instrument.

Given these premises, it is relatively easy to understand the rationale for the practices articulated initially in the provisions of the Ottawa Declaration and refined through the subsequent workings of the

Arctic Council. The members of the council are Canada, Russia, the United States and the five Nordic states described as "the Arctic States." There are no provisions for accepting additional members, though the fact that the Ottawa Declaration is not a legally binding instrument means legally that it could be restructured or even replaced through the adoption of a new ministerial declaration with different provisions. As widely noted, the designation of Indigenous peoples' organizations as Permanent Participants to "provide for active participation and full consultation with the Arctic indigenous representatives within the Arctic Council" constitutes an innovation that would have been hard to incorporate in the terms of an international legally binding instrument [1]. All others—non-Arctic states, intergovernmental organizations, and nongovernmental organizations—are relegated to the status of Observers. As the Ottawa Declaration indicates and as the practice of the council has made clear, the member states control decisions regarding all rules of procedure relating to the activities of the Observers [13].

Given the membership of the Arctic Council, it makes sense that the spatial scope of the council's activities extends to the circumpolar Arctic. But there are two interesting complications regarding this provision pertaining to spatial coverage. One arises from a clear distinction between the Western Arctic and the Eurasian Arctic regarding the southern boundary of the region. In the Western Arctic, the council's remit extends southward to 60°N and even a bit further in the case of Alaska. A similar boundary in the Eurasian Arctic would pass close to Oslo, Stockholm, Helsinki, and St. Petersburg, an arrangement unacceptable to decisionmakers in these realms. Although this is partly a simple matter arising from the configuration of land masses in the Northern Hemisphere, it introduces a distinct asymmetry into the constitutive provisions of the Arctic Council. In addition, many issues of environmental protection and sustainable development in the Arctic involve actions taking place outside the region. This is true whether we are concerned with matters relating to persistent organic pollutants and heavy metals or to matters arising from the extraction of hydrocarbons in the region. As a result, there is an arbitrary element to any effort to draw a clear spatial distinction between the Arctic region and the outside world.

The substantive remit of the Arctic Council articulated in the Ottawa Declaration flows directly from the idea of the Arctic as a low-tension region that merits treatment as a zone of peace. The council is to promote "cooperation, coordination and interaction among the Arctic states ... on common Arctic issues, in particular issues of sustainable development and environmental protection in the Arctic" [1]. Addressing environmental protection was easy, since the council inherited the fully operational working groups of the AEPS. Sustainable development presents a greater challenge. Not only is environmental protection generally regarded as one of the pillars of sustainable development; the reach of sustainable development into the spheres of economic prosperity and sociocultural well-being is also anything but clear. Still, it is clear that the founders of the Arctic Council meant to direct attention toward the domain of low politics and to insulate the council from complications associated with efforts to address matters of high politics. An interesting artifact of this practice involves the idea that the Arctic Council sometimes can serve as a forum for constructive contacts among representatives of states (e.g., Canada, Russia, the United States) that are engaged simultaneously in relatively sharp conflicts in other domains.

## 3. The Challenges of the 21st Century

Such are the premises and practices underlying the regime the Arctic states put in place during the 1990s. Whatever the suitability of this architecture for a regime addressing needs for governance at that time, we need to ask hard questions about the fit between this architecture and the challenges arising in the Arctic today and likely to come into focus during the foreseeable future. A straightforward way to proceed is to revisit the premises and practices described above in the light of both current developments in the Arctic itself and the evolution of links between the Arctic and the outside world.

Some have argued that the narrative of the circumpolar Arctic as a distinctive region with a policy agenda of its own was never persuasive. They point to sharp differences in the political

histories of the North American Arctic, Fennoscandia, and the Russian Arctic as evidence for this observation and conclude that the concept of the circumpolar Arctic as an international region is a flawed construct [14,15]. Be that as it may, there are good reasons today to question both the distinctiveness of the Arctic as an international region and the proposition that the Arctic is a low-tension region that fits the description of a zone of peace.

Above all, the impacts of climate change, a global development rooted in the activities of industrial societies located in the mid-latitudes, have emerged as dominant forces in the circumpolar Arctic. Feedback processes triggered by these impacts are amplifying the effects of climate change on a global scale. In biophysical terms, there is compelling evidence that the Arctic is experiencing what scientists call a bifurcation or a state change giving rise to what observers now describe as the 'new' Arctic. This state change is generating profound consequences both for the human residents of the Arctic itself and for outside actors who have begun to think about the Arctic through a new lens. Each of these developments deserves the attention of those concerned about meeting the challenges of Arctic governance in the 21st century.

Increasingly, Arctic residents and their communities are facing challenges driven by the global forces of climate change. These include coastal erosion producing pressure to relocate communities, melting permafrost causing increasingly severe disruption of infrastructure, ecological changes affecting the availability of both marine and terrestrial mammals, widespread fires impeding normal activities in sizable areas, and volatile weather patterns complicating day-to-day decisions about the conduct of subsistence practices. While Arctic residents can endeavor to inform policymakers operating in global arenas (e.g., the UNFCCC COPs) about these dramatic developments, they must find ways to adapt to the impacts of these global forces in securing their own well-being. At the same time, scientists have documented the importance of Arctic feedback mechanisms (e.g., increased absorption of solar radiation following the melting of sea ice, release of methane associated with melting permafrost) in accelerating the pace of climate change on a global scale [16]. With regard to climate change, the Arctic is ground zero on a global scale. In this respect, regional and global concerns have merged into an integrated policy agenda [17].

Similar remarks are in order regarding issues arising from increased access to the natural resources of the Arctic under the conditions associated with the 'new' Arctic. While all estimates need to be treated with a healthy dose of caution, the Arctic certainly contains a sizable fraction of the world's recoverable reserves of hydrocarbons as well as a variety of minerals ranging from staples like lead, zinc and iron ore to more esoteric resources like rare earths. The prospect of developing these resources is appealing both to economic decisionmakers and public policymakers within the Arctic and beyond. The government of Russia has accorded top priority to the exploitation of Arctic hydrocarbons in its drive to reestablish the country as a great power capable of playing a prominent role on a global scale. Chinese leaders have identified the Arctic as a component of the Belt and Road Initiative designed to create a far-flung network of coordinated economic relationships expected to underpin China's global strategy as an emerging superpower. Chinese, Dutch, and French corporations, as well as corporations based in the Arctic states, are among the leaders in initiatives aimed at the extraction of Arctic resources on a large scale.

The profitability of these economic initiatives is subject to global market forces and global policy actions. Arctic resources are expensive to produce and to ship to southern markets. Fluctuations in world market prices can make even large deposits of oil and gas in the Arctic (e.g., the supergiant Sthokman gas field in the Russian segment of the Barents Sea) unprofitable to develop. Any serious effort to come to terms with the global problem of climate change would make Arctic hydrocarbons less and less attractive in both economic and political terms. Advocates of vigorous action to address climate change argue that Arctic hydrocarbons must remain untapped. In short, the Arctic agenda and the global agenda have merged with respect to matters of political economy as well as the challenge of climate change.

An important consequence of these developments is that the Arctic is once again subject to the interplay of high politics. This does not mean we are witnessing the emergence of a new 'great game' in the Arctic or that the circumpolar Arctic is likely to become the scene of armed clashes during the foreseeable future. Yet, the Arctic is critical to Russia's strategy for reasserting its global status as a great power. China has made clear its intention to include the Arctic in its strategy for exercising its influence as an emerging superpower. Reacting defensively, the United States is making assertions about the need to protect its interests in the Arctic, even as its attention is focused on hot spots in other parts of the world including Iran, North Korea, and Venezuela.

Regarding Arctic governance, two important observations emerge from this account. One is that the Arctic is no longer a peripheral region with a policy agenda centered exclusively on issues of environmental protection and sustainable development that can be addressed largely in regional terms. The other related observation is that major non-Arctic states like China and intergovernmental organizations like the European Union are no longer willing to be content with the role of Observers in the Arctic Council when it comes to the pursuit of their Arctic interests.

An obvious implication of these developments is that they raise questions about the Arctic states' claim to paramountcy regarding Arctic issues based on their "sovereignty, sovereign rights, and jurisdiction." No one denies the special place of the Arctic states regarding Arctic issues, though much of the territory of all these states, with the exception of Iceland, lies outside the Arctic, and the major drivers of public policy in these states are not Arctic in character. In fact, most of the non-Arctic states interested in the Arctic make a point of professing respect for the interests of the Arctic states in the region. However, these states also are making expansive claims to being legitimate stakeholders in the Arctic and in China's case, to being a "near-Arctic state." Combined with the restrictiveness of the role of Observer in the Arctic Council, the effect of this situation is to encourage non-Arctic states to look to avenues to pursue their Arctic interests that offer more scope for the fulfillment of their goals. These include the negotiation of bilateral agreements pertaining to matters of mutual interest (e.g., China's role in the development of the Yamal LNG project and the associated port of Sabettta in the Russian Arctic), participation in multilateral forums that are more welcoming than the Arctic Council (e.g., the annual Arctic Circle event held in Reykjavik, Iceland), and engagement in intergovernmental agreements relating to Arctic issues not developed under the auspices of the Arctic Council (e.g., the Central Arctic Ocean fisheries agreement). The overall effect of these developments is to raise serious questions about the characterization of the Arctic Council as the "preeminent high-level forum of the Arctic region." A particularly striking development in this regard is the failure of the council, for the first time ever and due largely to American opposition to any reference to climate change, to reach agreement on the terms of a Ministerial Declaration at its 7 May 2019 meeting in Rovaniemi, marking the close of the Finnish Chairmanship.

These observations do not call into question the proposition that the Arctic is not a vacuum when it comes to the establishment and operation of arrangements designed to address needs for governance. We are witnessing the creation of a growing complex of arrangements dealing with Arctic issues. The terms of some of these arrangements have been worked out under the auspices of the Arctic Council, though the council itself is an informal body lacking the legal authority to adopt substantive agreements. These include the legally binding agreements on search and rescue (2011), oil spill preparedness and response (2013), and the enhancement of scientific cooperation (2017). The formal parties to all these agreements are the eight Arctic states. However, what is more striking is the development of governance arrangements that are not linked to the Arctic Council and that allow for the participation of non-Arctic states as members. Among the most significant are the Polar Code adopted by the International Maritime Organization to address issues relating to commercial shipping in the Arctic (2014/2015); the forum of science ministers from countries interested in Arctic research (initiated in 2016), and the regime dealing with fisheries in the Central Arctic Ocean (2018). As the 2015 Conference on Global Leadership in the Arctic: Innovation, Engagement, and Resilience (GLACIER), bringing together officials from over 20 countries in an event organized by the Obama Administration

in the United States in preparation for the negotiation of the Paris Climate Agreement in December of that year, makes clear, Arctic developments also have acquired a prominent place in efforts to respond to needs for governance relating to climate on a global scale.

Thus, we are witnessing a proliferation of regimes dealing with a range of needs for governance in the Arctic or relating to matters affecting the Arctic in important ways. What is less clear is what to make of the resultant assemblage of arrangements. Are we witnessing increasing institutional fragmentation that will have the effect of detracting from the performance of efforts to address specific needs for governance? Are there opportunities to manage the resultant complex in such a way that the whole is greater than the sum of the parts? What is the appropriate role for the Arctic Council? Is there a need to make more or less significant adjustments in significant features of the Arctic Council to allow it to perform effectively in this role?

## 4. Managing the Arctic Regime Complex

The rapidly growing literature on international governance encompasses several lines of analysis that may help in addressing these questions. Within the Earth System Governance community, there is an extensive literature on what analysts call institutional fragmentation [18]. There are many situations in which multiple regimes deal with matters arising in the same issue area or spatial area in the absence of any well-defined mechanism for integrating or at least coordinating their activities. The implication embedded in characterizing these situations as fragmented is that this an undesirable condition from the perspective of effective governance and that finding ways to reduce fragmentation should be a priority objective in any effort to improve the performance of governance systems. The classic recipe for pursuing this goal is to negotiate a comprehensive treaty (e.g., the Stockholm Convention on persistent organic pollutants) or to create an umbrella convention within which to nest a variety of linked protocols dealing with specific issues. A familiar example is the 1979 Convention on Long-Range Transboundary Air Pollution [19], which has a number of protocols dealing with specific pollutants such as sulfur dioxide, nitrogen oxides, and volatile organic compounds.

Conditions prevailing in the Arctic, however, suggest this is not a suitable strategy for coming to terms with the challenges of the 21st century. Numerous proposals for the development of an Arctic Treaty in the 2000s failed to gain traction; the debate about these proposals made it clear that an Arctic Treaty would have a number of drawbacks, even in the unlikely event that the relevant players agreed to enter into such a compact [20]. Fortunately, the literature on international governance suggests another way to think about the operation of multiple regimes that seems more promising as an approach to Arctic governance. The key concept here is the idea of a regime complex treated as a collection of distinct institutional arrangements dealing with related matters but not organized into a hierarchical structure [21]. Such complexes may deal with identifiable issue areas (e.g., the regime complexes for plant genetic resources and for climate) or with spatial areas (e.g., the regime complex for Antarctica) [22–24].

Research on regime complexes has produced two major findings of interest in this discussion of Arctic governance. One is that interactions between or among the individual elements of a regime complex need not give rise to conflicts that are difficult to resolve. Analysts have concluded both that these interactions often proceed without generating conflicts or tensions among distinct elements and that such interactions sometimes are synergistic in the sense that the various elements work together to address needs for governance in ways that would not be possible in the absence of such interactions [25]. The other significant finding is that it is possible, under some conditions, to manage interactions among the elements of regime complexes in ways that not only alleviate the danger of conflicts but also enhance the capacity of the complexes to meet needs for governance arising in the relevant issue area or spatial area [26]. An awareness of this prospect among those responsible for designing and administering individual elements can help.

So, there is no need for immediate alarm in response to the observation that we are witnessing a proliferation of issue-specific arrangements dealing with a collection of Arctic or, in some cases, polar

issues. These arrangements fall into two broad categories. Some focus on the governance of human activities occurring within the Arctic itself. The arrangements dealing with polar bears, search and rescue, oil spill preparedness and response, commercial shipping, the Central Arctic Ocean fisheries, Arctic marine-protected areas, and the conduct of science in the Arctic all belong to this category. The other category includes regimes that are global in scope but deal with issues that are of intense interest to those concerned with the well-being of the Arctic's human residents and biophysical systems. The international regimes dealing with ozone-depleting substances, persistent organic pollutants, heavy metals, greenhouse gases, and highly migratory species of birds, fish, and great whales all belong to this category. Any effort to assess the performance of the Arctic regime complex must cover elements belonging to both broad categories [27].

Yet, the two types of cases not only differ in terms of architecture, they also present different challenges for those concerned with Arctic governance. In the first category, the challenge is to make sure that the individual elements of the complex do not interfere with one another. For example, it is obviously important to ensure that shipping is managed in such a way that it does not disrupt the migration of marine mammals or harm marine-protected areas. In the second category, the goal is to make sure that global deliberations are informed by observations regarding Arctic developments and are as responsive as possible to conditions prevailing in the Arctic [28]. For example, global efforts to come to terms with climate change need to be informed by credible evidence regarding the impacts of climate change in the Arctic and the nature of the feedback processes linking Arctic developments (e.g., the recession of sea ice, the melting of permafrost) to the global climate system.

What mechanisms are available to take on these managerial roles regarding the Arctic regime complex during the foreseeable future? The obvious answer involves looking to the contributions of the Arctic Council either in its current configuration or in some revised form [29]. In thinking about the activities of the council, it is important to draw a clear distinction between two distinct approaches. One approach treats the Arctic Council as the voice of the Arctic states. On this account, the basic purpose of the council is to defend the primacy of these states in the realm of Arctic governance based on their "sovereignty, sovereign interests and jurisdiction" in the region. This approach amounts to circling the wagons, emphasizing solidarity among the eight Arctic states and holding the line against rising interests in Arctic issues on the part of non-Arctic states and other influential actors like the European Union. The other approach looks to the Arctic Council to play an important managerial role relating to the Arctic governance complex as a whole, including arrangements in which a variety of non-Arctic states and nonstate actors are active participants. Maximizing the effectiveness of the council in playing this role would require adjusting some of the constitutive provisions set forth in the Ottawa Declaration, a point I return to below.

Which of these options makes the most sense under the conditions prevailing today and likely to arise during the foreseeable future? There is no correct answer to this question. The desire of the Arctic states to protect their position of primacy regarding the treatment of Arctic issues is easy enough to understand. Considering their physical location and their longstanding interest in issues arising in the circumpolar Arctic, the clear sense among policymakers in the eight Arctic states that the rest of the world should acknowledge their primacy in the region is understandable. However, the critical question is whether hanging on to this position in the light of ongoing developments in the world at large, as well as in the Arctic more specifically, is realistic as an approach to Arctic governance going forward. More specifically, will a policy of using the Arctic Council as a bulwark against pressures emanating from non-Arctic states prove successful in holding the line or will it fail to produce the desired results and, at the same time, erode the legitimacy of the council?

My own sense is that we should be thinking hard at this stage about possible adjustments in the constitutive provisions of the Arctic Council that would enhance its capacity to manage the evolving Arctic regime complex in an effective manner. The good news in this regard is that these provisions are set forth in a ministerial declaration rather than an international legally binding instrument. Any changes all parties deem both politically desirable and practically feasible could be introduced in the

form of a new ministerial declaration superseding (elements of) the Ottawa Declaration without any need to go through the laborious process of amending an existing treaty and waiting for new provisions to enter into force. In today's world of complex and dynamic systems, this is a substantial advantage. It allows us to avoid lock-in regarding the various elements of the Arctic governance system. This also means that any revisions in the constitutive provisions of the Arctic Council the parties were to adopt at this stage could be revisited again in the light of continuing changes in needs for governance and adjusted in a comparatively easy manner.

The real question, then, is whether it is possible to reach consensus in political terms on adjustments in the constitutive provisions of the Arctic Council. In my judgment, any serious effort to address this matter must focus on two issues: membership in the council and the framing of the council's remit. Taking the question of membership first, a simple solution would be to expand the category of members of the council, much as the Antarctic Treaty System has done in adopting a relaxed interpretation of the provisions regarding qualifications for membership in Article 9 of the Antarctic Treaty to enlarge the group of states accepted as Antarctic Treaty Consultative Parties. However, any proposal along these lines would encounter determined opposition from existing Arctic Council members and most likely from the Permanent Participants. This suggests the need to develop more innovative approaches to the issue of membership in the council. For example, there may be a useful distinction between terrestrial issues of interest mainly to the eight Arctic states and marine and atmospheric issues of interest to a larger membership. There may be room for creating a bicameral system in which there is some recognized division of authority between a council open to a broader membership and a regional board whose members are the eight Arctic states. Or it may be possible to devise a system of weighted membership based on some measure of contribution to the work of the council, as in the case of the World Bank. The point is not to promote the adoption of any one of these options. Rather, it would make good sense to devote time and energy to developing innovative approaches to membership that would make sense in enhancing the capacity of the Arctic Council to manage the Arctic regime complex in the coming years, without running into intense opposition from any major stakeholder groups including the Permanent Participants.

In some respects, adjusting the Arctic Council's remit strikes me as an easier (though still complex) issue to address. The Ottawa Declaration directs attention to matters of environmental protection and sustainable development, while saying explicitly that the council should not deal with matters of military security. The obvious motivation underlying this provision was a desire on the part of the signatories to focus on matters of low politics in the interests of maintaining the Arctic as a zone of peace and cooperation [30]. Understandable at the time, it turns out to be impossible to adhere to this formulation as a matter of practice in today's world. In part, this has to do with the fact that sustainable development is an inclusive category, which subsumes environmental protection and encompasses the vast array of other concerns included within the UN's Sustainable Development Goals. Partly, the difficulty arises from the fact that with the reemergence of Russia as a great power, the rising interest in the Arctic on the part of China as an emerging superpower, and the defensive posture the United States has adopted, high politics have returned to the Arctic. As I have already indicated, this does not mean that we should expect the occurrence of armed clashes in the circumpolar Arctic anytime soon. However, it is apparent that efforts to address a wide range of Arctic issues now have a political dimension they lacked before and that this aspect of the international relations of the Arctic is destined to become even more salient in the coming years.

To some degree, the Arctic Council has begun to adapt to this development in practice, though it has avoided any formal recognition of such a shift. The creation of related bodies like the Arctic Economic Council and the Arctic Coast Guard Forum constitutes a significant development in the practices of the council implicitly if not explicitly. This is not a bad thing, though it might make sense at some stage to acknowledge this development explicitly and to consider its implications systematically rather than making believe publicly that no such adjustments are being made. One helpful step would be to acknowledge that sustainable development is an overarching concern encompassing the three

pillars of environmental integrity, economic prosperity, and sociocultural wellbeing and to take steps to adjust or even reorganize the council's activities on the basis of this conceptualization of its remit.

Making adjustments in the constitutive provisions of the Arctic Council could enhance the capacity of the council to manage the Arctic regime complex in several significant ways. In concrete terms, it would open-up opportunities to make progress regarding issues that have stumped the council in recent years. A striking example involves the efforts of the Task Force on Arctic Marine Cooperation (TFAMC), created by the council in 2015 at the start of the US Chairmanship and extended in 2017 at the start of the Finnish Chairmanship to develop a comprehensive and coherent approach to marine issues in the Arctic. To be frank, the TFAMC has failed. The reasons are not difficult to identify. In its current configuration, the remit of the council is not broad enough to tackle an issue involving areas located outside the jurisdiction of the Arctic states (e.g., the Central Arctic Ocean), as well as the activities of organizations that are not subject to supervision on the part of the Arctic Council (e.g., the International Maritime Organization). This means the TFAMC was doomed from the outset. But this does not mean the relevant issues have gone away. A reconfigured Arctic Council with a more explicit remit to operate as the manager/coordinator of the Arctic regime complex might fare better in addressing this issue. The key to the role of the council under this scenario would be an emphasis on facilitating interactions and even promoting synergistic interplay among the various elements of the Arctic regime complex concerned with marine issues rather than attempting to come to terms with the issue on its own.

More broadly, a reconfigured Arctic Council could focus on the challenge of developing a new narrative relating to Arctic governance to adjust or replace the Arctic zone of peace narrative that played an important role in earlier years. To be clear, we should still be emphasizing peaceful engagement and coordination in addressing needs for governance in the Arctic. However, conditions today differ in important respects from conditions prevailing in the 1990s. The Arctic agenda and the global agenda have merged regarding a range of critical issues. High politics have returned to the interactions among major players in the Arctic.

What might become major premises of a new narrative? It is not the purpose of this article to provide an answer to this question. Any useful answer should emerge from a concerted effort involving both practitioners and analysts to consider a range of possibilities. Nevertheless, it seems clear already that a suitable narrative to undergird the next stage in the evolution of Arctic governance should emphasize the importance of paying attention to the idea of stewardship in orchestrating efforts to maintain the integrity of the Arctic's biophysical, economic and cultural systems [31]; the need to devise creative ways to handle interactions between the Arctic and the global system, and the usefulness of approaching governance as a matter of managing the Arctic regime complex rather than endeavoring to negotiate a comprehensive Arctic treaty.

## 5. Conclusions

My answer to the question posed in the title of this article is affirmative. As someone who was present at the creation of the Arctic Council and who has followed the work of the council closely over the intervening 25 years, I have no hesitancy in concluding that the council has performed well during this period. However, times have changed; the premises on which the council was founded have eroded and will continue to erode during the near future. The members of the council could endeavor to make use of this body to hold the line against the impact of the changes now unfolding. This would mean resisting adjustments in the constitutive provisions of the council and making an effort to use the council as a bulwark, supporting claims to primacy based on the assertion of "sovereignty, sovereign interests, and jurisdiction" on the part of the Arctic states. In my judgment, this strategy is not only likely to prove unsuccessful as a means of resisting the pressure of outside forces affecting the Arctic; it is also likely to fail as an approach to managing the increasingly complex collection of governance arrangements relating to the Arctic in an effective manner.

It would be better, I believe, to take seriously the issues arising when we focus on possible adjustments in the constitutive provisions of the Arctic Council relating both to membership and to the characterization of the council's remit. Any effort to achieve success in this endeavor would face an array of complex challenges. Given the return of high politics to the Arctic, specific efforts to come to terms with these challenges could well fail. This makes it critical to avoid naive expectations regarding what can be accomplished in this realm over a limited period of time. Still, this is not an excuse for choosing to avoid the issue or assuming that efforts to promote constructive reforms in Arctic governance are bound to fail. Social institutions that perform the function of governance are not meant to be treated as sacred constructs to be preserved unchanged in the face of all pressures arising from major alterations in the political, socioeconomic and biophysical settings within which they are embedded. Restructuring the terms of governance systems at the first sign of adversity is not a virtue. However, refusing to consider adjustments in the face of profound changes is not a recipe for success.

**Funding:** The Pan-Arctic Options Project, funded under NSF/Belmont Forum Award No. 1660449, supported the preparation of this article.

**Conflicts of Interest:** The author declares no conflict of interest.

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
