# Peer review of "Is It Time for a Reset in Arctic Governance?"

_sustainability, doi:10.3390/su11164497_

Round 1

Reviewer 1 Report

Review of “Is it time for a reset in Arctic Governance?”Manuscript No. 576556

I greatly appreciate the opportunity to review this manuscript.  There are few, but this author, who has the “long-view” perspective and personal involvement in the history of the Arctic Council and the Arctic region, who could offer such insightful and meaningful analysis of the governance of the Arctic, past and future.  The topic of this paper and the perspectives offered, therefore, are both timely and likely to be critically important as regards future effective governance of this region.  Clearly, the world - and the Arctic - has moved on from the time when the Arctic Council was established, and adapting to this new reality is both warranted and necessary.  If nothing else, this is the most important message put forward in the work submitted.  There should be, as the paper suggests, an inclusive, deliberative process to address how this adaptation should occur and what this new governance regime might look like when all the necessary discussion concludes.  If the ideas put forward in the paper help to begin the process, this would be a significant contribution.  As a respected member of the Arctic governance community, the author’s words possess considerable authority, and I suspect that this would be a paper immediately read by others similarly interested in the future of the Arctic and our attempts to effectively address the governance issues confronting us there. The paper as submitted would be, in my opinion, a significant contribution to the Arctic governance literature, and this would be my recommendation.  Beyond this, I offer some collegial ruminations on the challenges of successfully accomplishing this recommended path forward for the author’s consideration.   

Recognizing the potential significance of the central “message” of the paper (as I have interpreted it), I would observe that institutions like the Arctic Council, established and operating for around 25 years, become less adaptable over time, and develop institutional cultures that resist change and innovation.  US Supreme Court Justice William O. Douglas once said something to the effect that every agency should be abolished after ten years, as at that point, they begin to care more about their image than their mission. I have experienced this inertia personally in some of my interactions with the Council.  The constituent Arctic states’ pursuit of “sovereignty, sovereign rights, and jurisdiction” has only intensified over this same period, as the stakes are higher given the expanding accessibility to economic resources, both extractable commodities and valuable access for human uses like tourism and shipping, that has resulted from the effects of climate change.  Although not specifically mentioned in the manuscript, I think it is clear that the ongoing efforts of the Arctic states to extend the seaward boundaries of their EEZ’s through the CLCS process is being driven by this pursuit of “sovereignty” (and resources), which is certain to change, perhaps considerably, the jurisdictional map of the Arctic.  We are now a world dominated by polarization (no pun intended), self-interest, and conflict rather than collaboration, in contrast to when the Arctic Council was established.  While I do not disagree with the idea that Arctic governance must adapt at this new reality to remain relevant and meaningful as “the preeminent high level forum for the Arctic region”, I wonder whether we are still capable of making this happen, or perhaps whether all the key actors, including the Arctic Council itself, really want things to change, having found something of a comfortable niche for the Council to occupy?  There are perhaps – and I don’t believe this is an overstatement - some who would see more chaos rather than cooperation as in their best interests. 

The current American government cannot even govern itself, and is only interested in the Arctic because of the Russian military expansion in the region, and the threat of rising Chinese influence (the US has never been, nor may never be, an “Arctic nation”, notwithstanding its geography).  While at least twice mentioned in the manuscript, the threat of military conflict in the region is unlikely.  However, all too frequent announcements are made by many of the Arctic states of new Naval vessels being constructed purpose-built for Arctic operations, and at least the US, Russia, and Canada all have policies and strategies related to military presence and operations in the Arctic.  Clearly, not all this investment is to expand their surveillance capabilities or simply enhance their constabulary role in enforcing national laws. Is the Arctic Council, or some future version we might envision, capable of continuing to be a collaborative, deliberative, and inclusive forum at a time when we are so polarized, seeing potential adversaries around every corner? 

Global diplomacy as conducted by the key players in the Arctic has become largely dominated by individual bilateral or limited multilateral agreements negotiated behind closed doors (which is the stated policy – at least according to Twitter - of the current US administration).  Even if we were to change the membership structure and remit of the Council, is the depth of commitment of the Arctic states, and the global players who seek a greater role, sufficient to find some common ground?  As the author rightly points out, the Council established at a time favorable to collaborative approaches to regional governance, the Arctic still largely perceived a remote and inaccessible place where the sovereign rights of the Arctic states were mostly respected as valid and controlling, and the tenure rights of the Indigenous peoples who live there, and their vision of an Arctic future, was being more widely recognized.  However, this is not exactly the reality we face today.  I would like to be more hopeful, but see what is ahead as a potentially insurmountable challenge.  While the paper does mention these challenges, appropriately framed in the discussion related to “institutional fragmentation”, I think perhaps there is more to be said about the implications of this new reality as some in the Arctic community seek to make the Council more relevant and effective.  There may be value in simply coming to terms with this increasing “fragmentation”, but perhaps attempting, in each agreement, to make it contribute less to this fragmentation (i.e. explicitly establishing appropriate coordination mechanisms within these various independent agreements, and also establishing similar explicit institutional coordination agreements, among the various independent governance entities that are party to these accords relevant to the Arctic region, for example).  Such a strategy might be pursued while the larger discussion is being conducted, but effectively achieving this may be as challenging as addressing the underlying issue of needed adaptation of the Council.     

The mountain up which we are rolling this boulder is steep, getting steeper, and those pushing the rock often cannot agree on how this task is best accomplished…and one wonders if all who are engaged really want to see the boulder reach the top.

Author Response

Reviewer 1 obviously is a well-informed and thoughtful observer of Arctic affairs. His/her observations deserve careful consideration.

Much of what the reviewer has to say takes the form of "collegial ruminations on the challenges of successfully accomplishing the recommended path forward." As such, they do not require a point-by-point response.

But I have made a concerted effort to address two points that come through in the course of these observations.

First, Reviewer 1 highlights the political changes in the Arctic that pose serious challenges for any effort to adjust or reform the architecture of the Arctic governance system. As a result, "the mountain up which we are rolling this boulder is steep." I do not regard this as a reason to avoid making the effort or even as a reason to become unduly pessimistic about the prospects for success. BUT the warning is well taken. I have revised several passages in the text of the article to indicate an awareness of these concerns and to ensure that the reader understands their importance in thinking about institutional reform regarding Arctic governance.

Second, Reviewer 1 has helpful things to say about making conscious efforts to build coordination mechanisms into the various elements of the expanding Arctic governance complex. I fully agree. It is not an easy matter to persuade those negotiating the terms of specific agreements to think about such matters. But it is definitely worth a try. My own view is that it is particularly important to ensure that those engaged in the work of the Arctic Council understand this point. I have made a number of revisions in the last major substantive part of the article to highlight this point, especially as it pertains to the work of the Council.

Reviewer 2 Report

Comments on “Is it time for a reset in Arctic Governance?” Young

It is somewhat unfortunate that this journal has an open review policy since the author of this submission is the world’s preeminent scholar in matters related to the Arctic in international relations and regime theory. One starts therefore with the premise that this latest submission will be publishable and it is just a question of how interesting the argument will be.

And it is an interesting argument and timely at that.

I have only a few comments to make.

Lines 155- 158. I don’t think that the claims of the Arctic Ocean coastal states depend on a few provisions of the LOSC (and the US of course is not a party to LOSC). I think that they base their claims more generally on the fact that the Arctic Ocean is subject to the law of the sea and that the law of the sea accords coastal states a preeminent position. Article 234 has some significance although its role will fade with the loss of sea ice and the IMO’s continuing work on issues of Arctic significance. Lines 281 – 305. I was struck by the lack of any references here and surprised to see Dutch and French corporations listed here as key Arctic actors (and no reference to Norwegian companies because Norway is part of the Arctic 8?) Line 299, sector? Line 303 untapped. Lines 294, 311 and elsewhere – some repetition re China. 343 – 348 While this deserves a mention I am not sure that it fits here; perhaps it should be earlier when addressing climate and the dissonance between the US and Europe on climate issues. Lines 433 – 458. I didn’t find the distinction between the two broad categories of arrangements to be particularly persuasive or useful; and I wonder what it adds to the paper? As for the way ahead I think that Young presents two realistic possibilities or scenarios. And I think it is correct to suggest that the membership issue is more difficult than the ‘remit’ issue. But I think that Young could be clearer on the remit issue. As a matter of practice the AC already has a very, very broad remit (anything except military security and fisheries); and I don’t think that Young is actually proposing that the AC’s remit should include military security. I recognize that Young does not want to answer the question as to the future major premise. So why then pick ‘stewardship’? Stewardship seems to me to be a word that is closely associated with Agreement on the Conservation of Polar Bears and the Ilulissat Declaration and the Arctic 5 (or 8) as stewards of the Arctic. How does that fit with an expanded membership? Two surprises. (a) I was surprised to see no mention of the Agreement on the Conservation of Polar Bears – which has been experiencing something of a revival and demonstrating that even if such an agreement cannot deal with large global change issues like climate change that have greatest effect on the health of polar bear populations there is still much that can be done by the range states. (b) while Young mentions the Permanent Participants earlier in the paper there is no mention of their role in the context of an expanded AC. It would be useful to have Young’s views on this issue.

Author Response

Reviewer 2 is complementary about my work on Arctic governance. Of course, I appreciate this.

Beyond this, he/she has comments about a number of specific passages in the text. In a few cases, these are just matters of fixing passages involving typos or inadvertent errors. I have fixed these.

But several more substantive issues arise. Here it what I have done:

First, the point about referring to the law of the sea in general and not just LOSC is correct. I have made the appropriate change.

Second, I have revised the text to make it clear that I do not mean to refer only to corporations based in non-Arctic states in contrast to corporations based in Arctic states.

Third, I have made adjustments to reduce the sense of repetitiveness regarding references to China. BUT I should add that Chinese activities are important in several contexts, so that it is important to refer to specific Chinese activities in a number of places.

Fourth, the two broad categories of arrangements are distinct, and the distinction between them is important to my argument. I have revised the text in an effort to make this clear.

Fifth, I have revised the text on the Arctic Council's remit to address the reviewer's comment. This was very helpful.

Sixth, I have retained the term "stewardship" in my reference to the premises of a new Arctic narrative. Since the work of Chapin et al. and others (including myself), this term is now in general circulation. But I have adjusted the text with the reviewer's comment in mind.  R

Seventh, regarding the two surprises: I have included a reference to the polar bear agreement and for that matter the 1920 Spitsbergen Treaty, and I have added comments regarding the views of the Permanent Participants in the later part of the article.